# Co-production of knowledge as part of a OneHealth approach to better control zoonotic diseases

Festus A. Asaaga[1]*, Juliette C. Young[2]*, Prashanth N. Srinivas[3], Tanya Seshadri[4,5], Meera A. Oommen[4], Mujeeb Rahman[4], Shivani K. Kiran[6], Gudadappa S. Kasabi[6], Darshan Narayanaswamy[6,7], Stefanie M. Schäfer[1], Sarah J. Burthe[8], Tom August[1], Mark Logie[1], Mudassar M. Chanda[9], Subhash L. Hoti[10], Abi T. Vanak[4,11,12], Bethan V. Purse[1]

1 UK Centre for Ecology & Hydrology, Wallingford, United Kingdom, 2 Agroécologie, INRAE, Institut Agro, Univ. Bourgogne, Univ. Bourgogne Franche-Comté Dijon, France, 3 Institute of Public Health, Bangalore, India, 4 Ashoka Trust for Research in Ecology and the Environment, Bengaluru, India, 5 Tribal Health Resource Center, Vivekananda Girijana Kalyana Kendra BR Hills, Bengaluru, India, 6 Department of Health and Family Welfare Services, Government of Karnataka, Shivamogga, India, 7 ICMR-National Institute for Traditional Medicine, Belgavi, Karnataka, India, 8 UK Centre for Ecology & Hydrology, Edinburgh, United Kingdom, 9 ICAR-National Institute of Veterinary Epidemiology and Disease Informatics, Ramagondanahalli, Yelahanka New Town, Bengaluru, Karnataka, India, 10 ICMR-Vector Control Research Centre, Puducherry, India, 11 DBT/Wellcome Trust India Alliance, Hyderabad, India, 12 School of Life Sciences, University of KwaZulu-Natal, Durban, South Africa

* fesasa@ceh.ac.uk (FAA); Juliette.Young@inrae.fr (JCY)

**Data Availability Statement:** Given the potential of identifying the individual who participated in our multi-stakeholder workshops and violating their confidentiality, our study dataset will not be shared

## Abstract

There is increased global and national attention on the need for effective strategies to control zoonotic diseases. Quick, effective action is, however, hampered by poor evidence-bases and limited coordination between stakeholders from relevant sectors such as public and animal health, wildlife and forestry sectors at different scales, who may not usually work together. The OneHealth approach recognises the value of cross-sectoral evaluation of human, animal and environmental health questions in an integrated, holistic and transdisciplinary manner to reduce disease impacts and/or mitigate risks. Co-production of knowledge is also widely advocated to improve the quality and acceptability of decision-making across sectors and may be particularly important when it comes to zoonoses. This paper brings together OneHealth and knowledge co-production and reflects on lessons learned for future OneHealth co-production processes by describing a process implemented to understand spill-over and identify disease control and mitigation strategies for a zoonotic disease in Southern India (Kyasanur Forest Disease). The co-production process aimed to develop a joint decision-support tool with stakeholders, and we complemented our approach with a simple retrospective theory of change on researcher expectations of the system-level outcomes of the co-production process. Our results highlight that while co-production in OneHealth is a difficult and resource intensive process, requiring regular iterative adjustments and flexibility, the beneficial outcomes justify its adoption. A key future aim should be to improve and evaluate the degree of inter-sectoral collaboration required to achieve the aims of OneHealth. We conclude by providing guidelines based on our experience to help funders and decision-makers support future co-production processes.

openly but may be available upon specific request. Researchers wishing to access the dataset used in this study should contact the UK Centre for Ecology & Hydrology Institutional Data Access contact via Jim Chiazzese (email: jimchi@ceh.ac. uk). The disease case data cannot be shared publicly due to Reasonable Security Practices and Procedures and Sensitive Personal Data or Information Rules enforced by Government of India. Researchers wishing to access the human outbreak data used in this study should contact the following officials at the Department of Health and Family Welfare Services, Government of Karnataka, Director (email: director-hfws@karnataka.gov.in), Joint Director of Communicable Diseases (CMD) (email: jdcmd-hfws@karnataka.gov.in).

**Funding:** The MonkeyFeverRisk project that led to these results is supported by the Global Challenges Research Fund and funded by the MRC, AHRC, BBSRC, ESRC and NERC [grant numbers MR/P024335/1 and MR/P024335/2], awarded to BVP, SLH, MVM, ATV, MMC, MAO, JCY, PNS, SJB and GK. PNS received support from the DBT/Wellcome Trust India Alliance Fellowship number IA/CPHI.16/1/502648. Additional support was provided to BVP and FAA from the NERC SUNRISE project [grant number NE/R000131/1]. The funders had no role in the study design, data collection and analysis, decision to publish, or preparation of the manuscript.

**Competing interests:** The authors declare that they have no competing interests.

## 1. Introduction

Zoonotic diseases that originate from animals make up 60% of emerging infectious disease events worldwide [1] and disproportionately affect poor tropical communities [2–4], accounting for an estimated 26% of Disability Adjusted Life-Years lost to infectious diseases in low- and middle-income countries (LMICs). Globally, a majority of poor communities depend on healthy ecosystems for livelihoods, welfare and food security particularly in LMICs. Upsurges in incidence of several high burden zoonoses have been linked to ecosystems being degraded or destroyed (e.g. malaria [5], Leishmaniasis [6,7], Crimean-Congo Haemorrhagic Fever Virus) while differential exposure rates among communities and individuals have been linked to specific human activities and resource-use within ecosystems [8–11].

Despite increased global and national attention on the need for effective strategies for zoonoses control, rapid action is often deterred by poor evidence-bases and limited coordination between stakeholders who do not normally act in concert (e.g. animal health, public health, forest departments), particularly in LMICs including India [12,13]. Such responses are also expected to be led by public health departments, which are typically organised vertically (i.e. responding to priorities and programs conceived higher up in their own sector rather than exchanging with other sectors). Health departments in many countries (especially in the global South) struggle with under-financing and the need to organise interventions for non-communicable diseases, maternal and child health and non-zoonotic human infectious diseases. In addition, the limited leadership in such departments may impede the effective control of zoonoses, both in terms of technical leadership (e.g. evidence-based disease control measures) and managerial leadership regarding the organisation of human and financial resources [14].

The global *OneHealth* paradigm is an approach which recognises the "interconnectedness of human health, wildlife and domestic animal health and the environment" and the value of evaluating these interactions in an integrated, cross-sectoral and transdisciplinary manner to reduce pathogen transmission risk and mitigate disease impacts [15–17]. Poor cross-sectoral integration has hampered effective policy-setting for zoonotic diseases and understanding of the mechanisms that underpin emergence, impacts and effectiveness of interventions [3,18,19]. Many of the current risk models and information systems for zoonoses are not tailored to the needs of decision-makers, at the most relevant scales, resulting in strategies to address zoonoses that are either inadequate, and/or not implemented by those who need them most [20].

Given such significant constraints, Leach & Scoones [20] recommend that disciplinary and sectoral perspectives should be "triangulated" with attention to framing assumptions, policy narratives, politics and values. Such a co-production approach is linked to a number of research strategies that aim to engage more and better with stakeholders, including Participatory Action Research [21], community-based participatory research [22] and Mode 2 knowledge production [23]. Knowledge production used in this context refers to the ways scientific knowledge is produced and applied. Alternatively referred to as transdisciplinarity as it affords the integration of knowledge from academic and non-academic stakeholders [24]. Two modes of knowledge production are commonly characterised—mode 1 and 2. Whereas in mode 1 problems are set and solved in a context governed largely by the interest of a specific community/discipline, mode 2 knowledge is carried out in a context of application through collaborative working by multidisciplinary teams [23,25]. Such methods actively engage stakeholders–seen as both knowledge holders and decision-makers–as agents of change in terms of the research process and its implementation [21,26–29].

Integrating different types of stakeholders to inform decisions has a strong grounding in international policy-making, enshrined in Article 8(j) of the United Nations Convention on

Biological Diversity, Agenda 21 of the Aarhus Conventions, the FAO/OIE/WHO collaboration (Tripartite) and associated regional or national approaches. The rationale for such integration arises from the hypothesis that such forms of integration encompass a broader diversity of knowledge, experience and expectations [30], leading to improved management [31,32]. In addition, collectively agreed decisions acknowledging diverse knowledge may be more socially and politically acceptable [33–35]. Besides, the integration of diverse knowledge on improved equity, stewardship and efficiency has been identified as being a priority for OneHealth initiatives [36]. Here we argue that co-production is a particularly useful approach to use in the context of developing acceptable and implementable strategies to better address zoonoses. This is a context where multiple stakeholders across public health, forestry, animal health sectors and scales (from local to national) and local communities have knowledge of, and knowledge needs relating to zoonoses, but where this knowledge is not integrated or communicated between stakeholders, with negative impacts on humans and their well-being. Thus, implementing a participatory co-production process (embedded within a OneHealth systems thinking) offers a practical lens to iteratively and collaboratively engage cross-sectoral stakeholders at different stages of the framing, development and implementation of context-specific knowledge and tools to improve zoonoses management. In essence, participatory co-production process approaches help appreciate and understand different viewpoints, priorities and decision-making processes and achieve better disease prioritisation, research, communication and implementation of existing data and tools [4,17,20,37,38].

Various authors have outlined the challenges of co-production processes [39–41]. Challenges include the paucity of examples of co-production around population-level policy and practice interventions, with co-production processes often focussing on single communities and not on policy change [42–44]. Co-production across sectors for zoonotic/communicable diseases also revolve primarily around prioritisation of diseases and research agendas, rather than longer term cooperation to co-develop and operationalise particular interventions or research studies [45,46]. Whilst some studies have highlighted the advantages of integrating stakeholder knowledge on key risk factors into epidemiological models to inform interventions [38], no studies have explored the approach of cross-sectoral and multi-scalar co-production in the development of understanding of risk factors, predictive models for zoonoses and control interventions. The potential outcomes and learning from such an approach could be far reaching for developing models and interventions for zoonoses in other settings.

In this paper, we outline a co-production initiative in southern India that implemented a novel interdisciplinary approach to create a decision support tool (DST) for control of Kyasanur Forest Disease (KFD) with stakeholders at local to national scales, across the public health, animal health and forest policy sectors. First, we introduce our study system and context and outline our co-production process, including the theory of change framework. We go on to describe the process of co-production, including the benefits and challenges encountered and how these were addressed through an iterative approach. We finish with a summary of lessons learned and recommendations for decision-makers and researchers for future co-production of zoonotic disease tools and management.

## 2. Methods and approaches

### 2.1 KFD study system

Kyasanur Forest Disease (KFD) is a tick-borne haemorrhagic viral disease that affects poor communities in the forests of the Western Ghats mountain range in Southern India (across the states of Karnataka, Maharashtra and Kerala), and can be fatal in up to 10% of unvaccinated infected people [47–50]. Communities at particular risk of exposure to KFD include,

forest dwellers, traditional societies and settled village communities that harvest non-timber forest products and access food, fuel and fodder, plantation and forest department workers, farmers who graze livestock or cultivate on forest fringes [47,48,51–53]. Aside from KFD affecting diverse forest users, it has a broad vector and host range, with a complex transmission cycle in which various tick species (notably the Haemaphysalis genus, but also Ixodes) and vertebrate hosts have been implicated, including wild rodents and shrews, monkeys and some birds [53]. Humans contract KFD when bitten by an infected tick but are incidental hosts for the disease and are not involved in onward transmission [54,55]. Therefore like the Lyme disease agent, Borrelia burgdorferi, KFD virus is a "spill-over pathogen" for which almost every human case represents a spill-over event from a wildlife reservoir via the infected tick vector [55].

The work reported here is a result of the *MonkeyFeverRisk* project, which aimed to develop an inter-disciplinary DST to help communities minimise exposure to zoonotic diseases whilst maximising the livelihood benefits derived from tropical forests. The project used a OneHealth approach and was interdisciplinary, linking expertise in public and animal health, forest and wildlife ecology, human behaviour and priorities, empirical measurements and models to understand the ecological and social processes that make communities more vulnerable to KFD.

**2.1.1 Actors involved in disease management in India.**   The organisational landscape for disease management in India is multi-layered involving different actors operating at different levels across national, state, district, Taluka (sub-district) and village/local government scales (Fig 1). Districts are the typical administrative sub-divisions in which disease management is organised, with sectorally-defined department heads who monitor and oversee operations within their sectors/departments. However, Indian districts are relatively large, covering populations of 1–3 million. Although organisation of animal and human health services is the mandate of sub-national state governments, disease surveillance is a function retained at the national government level under India's Constitution [13,56,57]. Among 'OneHealth' actors involved in disease management, four focal institutions have overarching responsibility for zoonoses prevention and control: the National Centre for Disease Control (NCDC), Indian Centre for Medical Research (ICMR) and the Indian Council for Agricultural Research (ICAR) and the national disease control programs implemented by the Ministry of Health and Family Welfare (for e.g. the National Vector-borne Diseases Control Program). The NCDC and ICMR are human health institutions under the Ministry of Health and Family Welfare (MoH&FW) focussing on human wellbeing. ICAR and the Department of Animal Husbandry and Dairying (AH&D) respectively fall under the Ministries of Agriculture and Farmers Welfare (MoA&FW) and Fisheries, Animal Husbandry & Dairying, promoting animal health for food production and safety. The NCDC provides technical guidance and support to state governments in the event of health emergencies and manages the respective national disease surveillance programme and the National Standing Committee of Zoonoses (NSCZ). The ICMR, an autonomous department within the MoH&FW (comprising 26 centres of excellence–e.g. National Institute of Virology, National Institute of Malaria Research), administers disease research through grants to respective medical colleges, universities and affiliated laboratories (see Fig 1). Conversely, ICAR regulates a large network of specialised research and teaching institutions focussed on agriculture and veterinary sciences. The Wildlife sector, which is the third 'OneHealth' sector [58], under the auspices of the Ministry of Environment, Forest and Climate Change (MoEF&C) responsible for wildlife health on conservation-related concerns [13,56]. The functionally disparate actors in disease management further highlights the importance of a co-production approach to better understanding and managing zoonotic diseases in India.

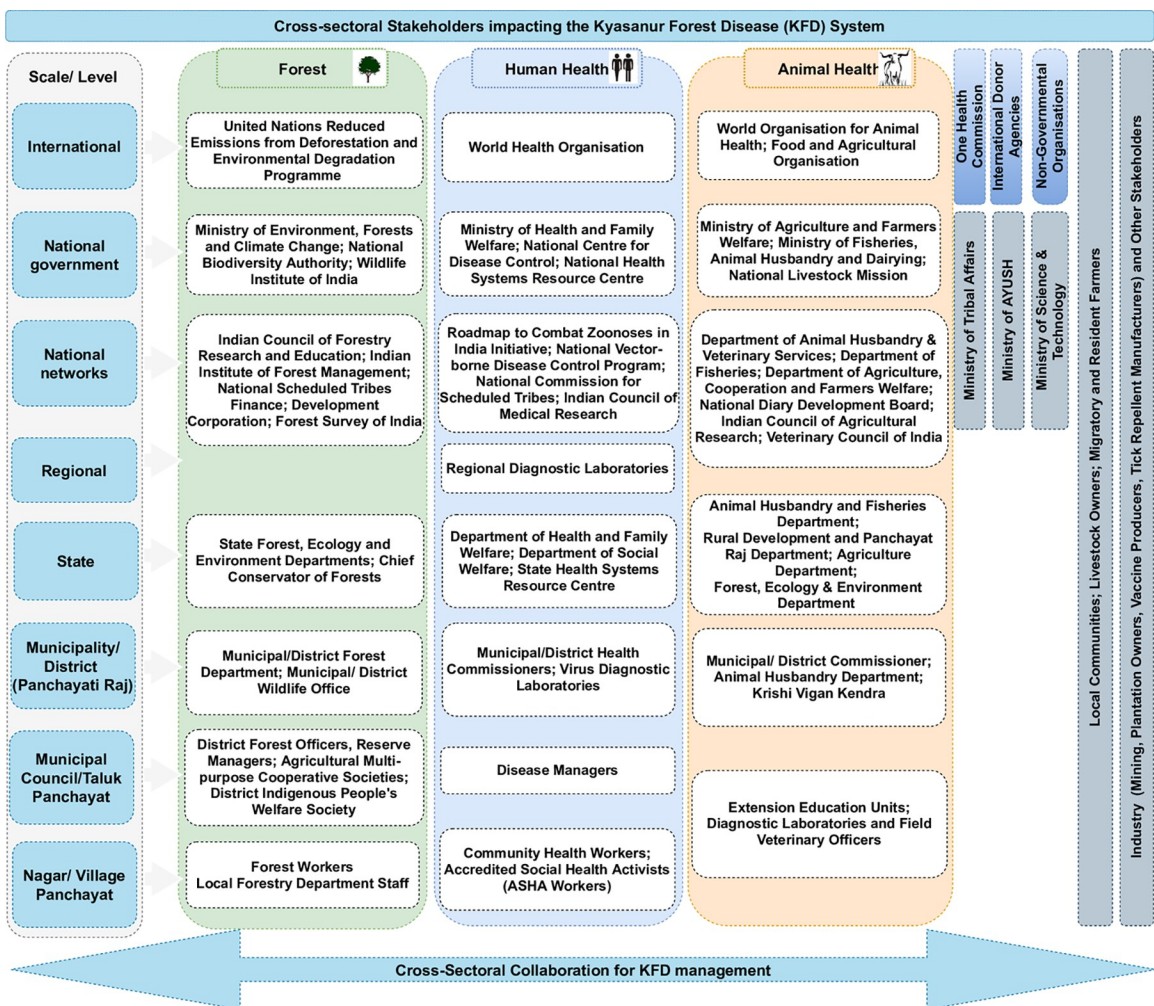

**Fig 1. A simplified illustration of the sectors and the politico-administrative actors of the KFD management in India including actors from each sector that are impinging on the system or are impacted by KFD in each sector.**

## 2.2 Co-production process

Co-production is particularly well suited to the development of models to understand and predict zoonotic diseases, where stakeholders from different sectors and scales are integral as knowledge holders and future model users. The steps of co-production in the context of the project were:

- Engaging with key stakeholders to frame the research

- Feeding in knowledge from wider stakeholders into research and resulting tools

- Validating the DST

**2.2.1 Selection of workshop participants.** *Prior to organising the workshops, the research team conducted a stakeholder* mapping exercise, which culminated in a list of current and potential cross-sectoral actors involved in KFD management at local, national or state level. Based on this stakeholder list, workshop participants were purposively selected to warrant (a balanced) representation of policy/decision-makers from the human health, animal health and

forestry sectors with designated roles in KFD management, who had the potential to provide relevant, detailed and diverse information about the KFD system. The variety of sectoral representation, participants' experience with KFD prevention and control, and the ability to assess the value of the DST and other project outputs were the main criteria for inclusion of participants.

In recruiting prospective participants for the workshops, formal introductory letters were sent to the appropriate state departmental directorates (in accordance with institutional protocols), explaining the objectives of the overall project, and requesting them to depute up to 7 staff from their respective departments to attend the workshops. The introduction letters were sent to the Karnataka state directorate of health and the NCDC (public health sector), Department of Animal Husbandry and Veterinary Services (DAHD) (animal health sector) and State Forest Department (environment sector) respectively. Two of the co-authors (MMC and SLH) followed up with telephone calls to further explain the purpose of the workshops and confirm participation at the sessions, in accordance with administrative protocol and local custom. The characteristics of the workshop participants are outlined in S1 Table (see supplementary material).

**2.2.2 Multi-stakeholder workshops.**   As part of the co-production process, a participatory consensus-building technique–nominal group technique (NGT)–was employed in the two multi-stakeholder workshops. The NGT is a stepwise, democratic and participatory process used to generate a list of collectively established priorities [59,60]. In this context, the first 'framing' workshop was convened in Bengaluru in August 2018, with 19 experts from different KFD-affected districts state level officials of Karnataka, Maharashtra and Kerala from the public health (6 participants) and animal health (10 participants), agriculture (one participant), forestry (one participant) and social welfare (one participant) sectors. Given the endemicity of KFD in a number of districts in Karnataka state, Bengaluru was deemed a convenient location for the workshops. The workshop aimed to map stakeholders' knowledge about KFD, prioritizing risk factors for the disease, identifying key policies that affect KFD, and feeding that knowledge into project approaches and models. To this end, the facilitated discussion and simple voting process (rank-ordering of risk factors) of the NGT helped circumvent typical power dynamics of the diverse stakeholder groups and afforded equitable contribution to group discussions [59]. Moreover, using the NGT during the workshop (1) afforded quick collation of considerable amount of information about stakeholders' priorities and perspectives within a short time frame using minimal resources in a single event, and (2) alleviated possible researchers' biases as data was interpreted by participants [59,60].

In between the framing workshop and the knowledge and experimentation workshop, the desk-based spatial DST was developed using a multi-pronged approach. Firstly, human case data were collated across the KFD-affected areas and modelled in relation to risk factors identified by stakeholders or in the scientific literature to produce and validate risk maps for disease spill-over (as described in detail in [61]. A desk-based decision support tool (based on the Shiny app interactive web technology from RStudio (https://shiny.rstudio.com/) was then constructed, integrating disease surveillance and landscape data, alongside the risk maps, accounting for the stakeholder preferences identified in the framing workshop for how seasonal and geographical information should be presented and overlaid to inform disease management. Importantly, a post-doctoral researcher from the research team was embedded in the health department throughout this process and, alongside data collation and modelling was involved in the KFD response, participating in interventions such as inter-departmental sensitisation workshops for decision-makers across sectors, and community awareness raising activities. This allowed for further reciprocal knowledge integration and feedback on utility and contextualisation of the developing project DST and risk guidance in between the project workshops.

The second workshop, focussing on knowledge integration and experimentation, was held in Bengaluru in December 2019. At this second workshop, we presented a prototype desk-based spatial DST to stakeholders (developed based on their input in the first workshop), encompassing predictive risk maps for human spill-over, maps of landscape risk factors and features that guide interventions, human cases and tick and monkey infections from surveillance, and awareness raising material on key risk factors and affected communities. As part of the experimentation process of the workshop, stakeholders were invited to test the DST. During this process, the stakeholders identified many functional characteristics to be enhanced or added to the DST. The project team divided these into categories (site- or area-based information, individual level information, temporal or alert-based information, awareness raising and capacity building information, and other) and identified time lines and feasibility of integration. We also scoped out whether development of a mobile App would benefit management and, if so, what functionality should be included and for which groups of end-users. The second workshop was attended by 34 practitioners and experts from different KFD-affected districts and state level officials of Karnataka, Maharashtra and Kerala from the public health (22 participants), animal health (6 participants) and other sectors (6 from agriculture, forestry and social welfare sectors). Despite efforts to achieve a balanced sectoral representation at both workshops, this was not achieved due to circumstances beyond the control of the project team. A case in point is the major flooding event which occurred in parts of Kerala and Karnataka states (coinciding with the first workshop), which resulted in the withdrawal of some invitees who had hitherto confirmed their participation. Besides, the first workshop clashed with a field programme of the forest department, which meant that some initially deputed officials had to decline the invitation to attend the workshop. The second workshop had a disproportionate representation of Public Health actors principally due to the specific expression of interest to participate received from the health departments officials stationed at Sagara (an area newly affected by KFD that season). The project team granted this request as it was deemed a good window of opportunity to obtain feedback on the prototype desk-based spatial DST as well as build good rapport for on-ward dissemination and uptake of project outputs. Nonetheless, further follow-up interviews and conversations with under-represented stakeholders (principally from the forestry department) during the course of the project (including the national KFD stakeholder consultation workshop) afforded in-depth understanding of their perspectives and inputs into the DST development.

In both workshops, the approach adopted was inclusive, building on small group discussions and developing joint worksheets. In both workshops, we allocated participants to sub-groups to ensure that each sub-group had representation of practitioners and experts from different districts and ideally from different sectors. A facilitator on each table moderated the discussion in line with the objectives. The facilitator was instructed to initiate the discussion with clear explanation of the session and discussion topics; to provide general instructions such as duration, ground rules and expectations of the discussion; and to note the important points for each topic on a worksheet. A rapporteur was allocated to each table, taking detailed notes from the discussions and supporting the facilitator. At the end of each discussion session, each rapporteur and facilitator communicated the main points to a colleague responsible for synthesizing the key points, who entered the information into the Mentimeter software to allow for prioritization of issues with workshop participants. The facilitators adopted an opened-ended and friendly approach, which ensured that the sessions were highly participatory, as participants were encouraged to feed in their knowledge and experience during the workshop using a range of different approaches including group discussions, brainstorming and prioritization exercises. Fig 2 presents a graphical representation of the retrospective theory of change on researcher expectations on the system-level outcomes of the co-production process adopted

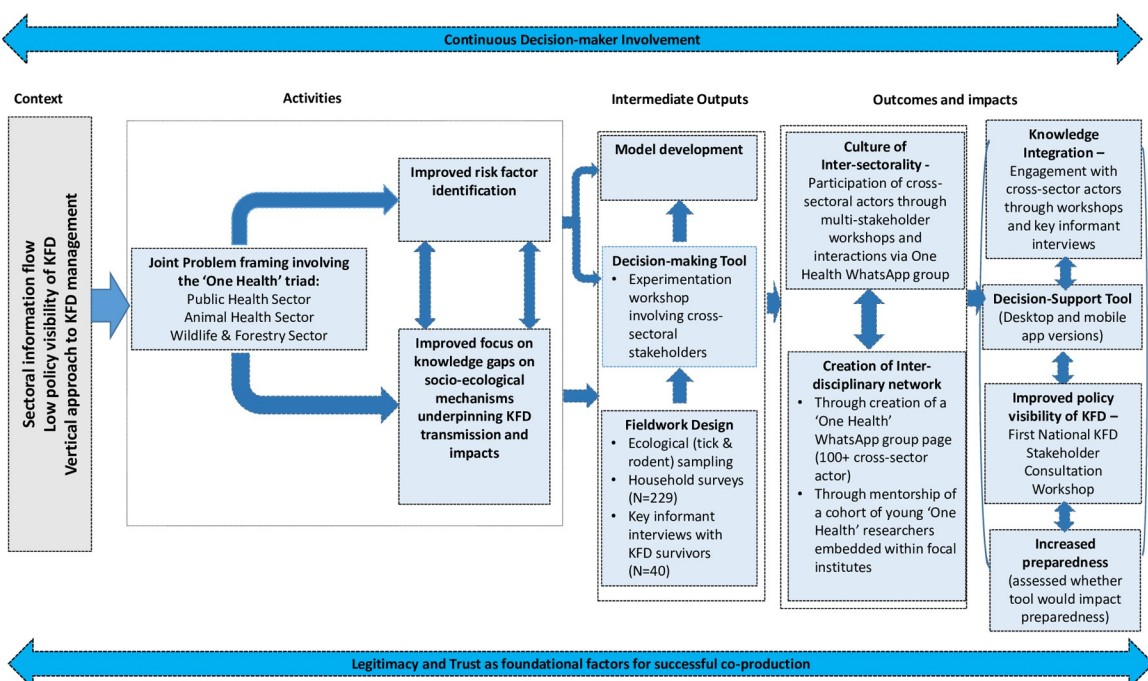

**Fig 2. Graphical representation of the retrospective theory of change on researcher expectations and the system-level outcomes of the co-production process adopted in the MonkeyFeverRisk project.** The key components of the theory of change framework were as follows: (i) planned activities which included joint problem-framing with cross-sectoral actors to better understand the contextual risk factors and knowledge gaps, (ii) intermediate outputs comprising the social and ecological fieldwork, design and experimentation of a decision-support tool, and (iii) anticipated outcomes which ultimately contributes to fostering inter-sectoral collaboration and increased disease preparedness.

in the context of our project. It highlights the various key elements and processes underpinning the co-production process and how they relate to the envisioned outcomes and impacts.

## 2.3 Data analysis

The collated qualitative data from the workshops (including audio transcriptions, observation notes and suggestions made by participants) were thematically analysed within a deductive framework. After transcribing the audio recordings of the workshop discussions, the transcripts were manually coded (line by line and axial) independently by three of the co-authors (JCY, MAO and FAA), following Braun & Clark's [62] thematic content analysis approach. Exemplar quotations from participants and observations of the facilitation team were also collated. Each reviewer (JCY, MAO and FAA) generated themes and sub-themes, which were subsequently compared and refined to produce the final thematic analysis. If there were any disagreements, reviewers discussed themes to collectively agree on revisions. All participant information were de-identified using pseudonyms for the analysis and report-writing; these pseudonyms are reported here.

## 2.4 Ethical approval and consent to participate

The study protocols were approved by the Institutional Ethics Committee of the Institute of Public Health (IPH IEC) (Study ID, IEC-FR/04/2017), Ashoka Trust for Research in Ecology and the Environment (IRB/CBC/0003/ATV/07/2018) in India, and received a Favourable Ethical Opinion from the Liverpool School of Tropical Medicine Research Ethics Committee in the United Kingdom (research protocol 17/062) under the MonkeyFeverRisk project. All

study participants were adults and gave full prior-informed verbal and written consent before the conduct of the workshops. The collated data from the workshops were duly anonymised using de-identifiers or pseudonyms prior to analysis.

# 3. Results

## 3.1 Insights, benefits and challenges from each stage of the co-production process

**3.1.1 Problem framing.** As part of the framing workshop, stakeholders were asked about what they perceived as key risk factors (see Fig 2). Facilitators compiled and typed up sticky notes (Part 1) for participants to rank individually. Participants were given 5 stars to use as they wished in ranking the identified risks. Based on the ranking exercise, the key risk factors that stakeholders felt needed to be addressed for effective KFD management are highlighted in Table 1. Whilst a number of these risk factors were addressed in the project (see last column of Table 1), it is worth noting that not all of the risk factors identified by stakeholders could be encompassed in models or in fieldwork (for a detailed description of the modelling approach employed, see Purse et al., 2020). For example, in spatial risk models it was not possible to integrate factors like variation in health data management or reporting of monkey deaths by the Forest Department (some species of primates are considered amplifying hosts for KFD), low vaccine uptake or awareness of KFD as model inputs [61], though these factors could be considered in model interpretation and factored into household surveys and information material.

**Table 1. Risk factors identified and ranked by participants of the framing workshop, and how these were integrated in the MonkeyFeverRisk project.**

| Ranking | Risk factors | Number of votes | How risks were addressed in project |
|---|---|---|---|
| 1 | Lack of education/awareness | 10 | Tick information cards were produced to inform local communities about risks from ticks and tick protection measures. Development of an educational video in progress. |
| 2 | Under or late reporting of monkey deaths | 9 | Accounted for in data interpretation in risk modelling |
| 2 | Deforestation and/or forest degradation | 9 | Integrated as a risk factor in models |
| 2 | Lack of awareness of preventative measures (tick repellants, vaccination) | 9 | Measured in cross-sectional household surveys WP2 Tick information cards produced (see above). |
| 3 | Lack of awareness or understanding of alternative hosts | 8 | Addressed in household and ecological surveys |
| 4 | Human use of forests | 7 | Addressed in household surveys and in spatial risk modelling |
| 4 | Low vaccination coverage | 7 | Addressed in household surveys and in spatial risk modelling |
| 4 | Poor diagnostics and surveillance | 7 | Improving surveillance and diagnostics is not a direct project aim but could result from a strengthened OneHealth network. Ecological analysis of vector and alternate hosts will inform surveillance strategies. |
| 4 | Lack of OneHealth policy | 7 | Project established a OneHealth WhatsApp network on KFD which facilitates networking amongst people involved in KKFD management. Project members attended National and State level technical committees on KFD and discussed OneHealth approach |
| 5 | Poor data management | 6 | The project provided a blueprint for future data management on KFD, for example ensuring that cases were georeferenced at a household level to capture landscape conditions favouring spill-over |
| 5 | Poor understanding of tick ecology | 6 | Addressed in ecological surveys |
| 6 | Side effects and concerns about vaccines | 5 | Measured as part of the household surveys but not a direct research project aim |
| 7 | Living in or around forests | 4 | Addressed in risk modelling, household surveys and ecological surveys (stratified by forest proximity) |
| 7 | Favorable environment for ticks | 4 | Addressed in ecological surveys (habitat associations were measured) |
| 7 | Poor tick identification | 4 | Addressed in ecological research and capacity building (see Table 2) |

A more complete list of ranked risk factors and what changed and/or strengthened over the course of the project is presented in S2 Table.

The participants of the workshop also shared what their key needs pertaining to KFD management were from the project (Table 2).

An important input from the framing workshop was the understanding from workshop participants of the wide range of interacting policies and actors that impact or are impacted by the disease system including longer term political and economic processes that might affect disease transmission. Concerning the impacts of national and state level policies on KFD management, key policies that were identified as having a negative impact or as being poorly

**Table 2. Key needs identified by participants of the framing workshop–and how these needs were addressed in the project.**

| Key needs identified by workshop participants | How needs were addressed in project |
|---|---|
| Human resources: need for better trained manpower; more equipment; tick experts and taxonomists | Institutional capacity for morphological and molecular tick identification was built in partner institutes and within the health system (training of district entomologists). Tick taxonomy resources were developed that will be made publicly available |
| Improved surveillance: need for active surveillance; surveillance for disease, vectors and hosts | WP3b provided risk maps and models that were integrated into a desk-based App "KFDExplorer" to improve targeting of surveillance. WP3a advanced understanding of the ecological communities most strongly linked to KFD and developed protocols for tick and small mammal surveillance. |
| Better diagnostic facilities | Not a direct research project aim but OneHealth network can advise on location/type of facilities |
| Better communication: real-time reporting; social media use | Part of experimentation phase |
| Funding for research and action | Not a direct research project aim but opportunities were communicated through the OneHealth network |
| Better understanding of disease ecology: alternative hosts and vectors; seasonality; tick movement; tick distribution; tick ID and taxonomy | Ecological surveys and research advanced this understanding, and produced Tick Information cards (see above). Published review of the ecological evidence base for current KFD management for disease managers [55]. |
| Vaccines and vaccination innovations: better quality/efficacy/single dose; availability; shelf life | Not a direct research project aim |
| Multi-sectoral coordination: better communication and coordination | Stakeholder workshops; WhatsApp groups, establishing a OneHealth network |
| Raise profile of KFD and hence generate political will for KFD control and management | Project members engaged with a wide range of media outlets to raise awareness of KFD and attended National and State level government technical committees on KFD to provide advice and describe the OneHealth approach |
| Improved knowledge, awareness and better practices for KFD management | Tick Information cards produced and video in progress–see above). Review of the ecological evidence base for current KFD management (see above). |
| Improve detection of at-risk human populations early | Ecological surveys and spatial risk models improve understanding of the landscape conditions favouring spill-over, whilst the household survey indicated livelihood risk factors and activities for KFD |
| Restrict human-forest interface wherever feasible | Covered in household surveys as part of raising awareness. Analysis of ecological data to identify important non-forest interfaces (other than forest) affecting human spill-over dynamics. |
| Remove invasive species | Ecological surveys measured links between invasive plants, tick abundance and KFD |

implemented were those concerning deforestation, grazing and encroachment in and around forest areas, with abrupt shifts in land use in these areas being identified as making communities more vulnerable to KFD. In the health sector, policy changes that were suggested to benefit management were making KFD a notifiable disease (discussed further at the 2019 National Technical Committee on KFD), learning from wide-scale vaccination and vector control campaigns for other diseases, improving the coordination between animal health professionals conducting monkey post-mortems and public health professionals involved in treating human patients, improving the screening of livestock for pathogens and ecto-parasites before transportation.

Concerning the mapping and forecasting information needed to manage KFD, workshop participants were of the opinion that making available risk predictions at scales from village-level to clusters of villages would be most helpful to plan vaccination and awareness campaigns. Participants requested that data on environmental risk factors such as climate, land use change, altitude and livestock densities should appear alongside risk maps in a tool, together with contextual features like roads and household locations that health managers routinely use to plan their management across the landscape [61]. Interfacing the seasonal activity of ticks with seasonal activities conducted by different groups of forest users in the tool was felt to be very important. Predictions of the month and villages at highest risk would be most useful at least six but better two months before the KFD season. The project team tailored the scale and appearance of the tools to these needs.

To summarise, the first workshop was instrumental in collaboratively framing the research priorities of the project and its future direction. As a direct result, the project approaches and models [61] were changed to reflect:

- New hypotheses, for example 'what makes communities more susceptible and exposed to KFD' that could then be tested with models/fieldwork within and beyond the project

- Improved integration of key risk factors into understanding and tools for zoonotic diseases

- Maintained focus on quantifying both ecological and social components of risk at regional and landscape scales

- Tuning the study grain and models to the scale of landscape use by people, hosts and vectors

- Developing predictive tools that account for the way that disease managers collect disease data, interpret and use seasonal and geographical information.

**3.1.2 Knowledge integration and experimentation.** Following the joint framing of the research, the next step in knowledge co-production focussed on knowledge integration and experimentation, in the second workshop (Fig 2).

As part of the knowledge integration process, a wide range of users and uses were identified by stakeholders (See Table 3). This identification was far wider ranging than we had initially planned in the project team, meaning more effort was needed to manage expectations about which users and uses could be encompassed in the project time frame, but this allowed for a tailoring of specific functionality to end-user needs. For example, stakeholders felt that the desk-based App would be suitable for officials involved in KFD management at block level up to National and State Levels. For users on the ground such as health workers and Primary Health Centre Medical officers, a Mobile Phone App was felt to be more suitable. Across the three main departments involved in KFD management, namely the Public Health, Animal Health and Forest Departments, different groups or levels of users that (a) might benefit from the DST (b) and/or might enter data into the DST were identified. For each user, Table 3

**Table 3. Key potential users and uses of the DST across the three main departments involved in KFD management, namely Public Health, Animal Health and Forest, as identified by stakeholders at the second workshop.**

| Level | Details | Animal Health Department | Public Health Department | Forest Department |
|---|---|---|---|---|
| National Level | User | | National Centre for Disease Control | |
| | Use of DST | | Guide supervision of Outbreak investigations, research planning, Technical meetings and reporting to higher National level officials | |
| State Level | User | Joint Director Animal Health | Joint Director Communicable Diseases | Principal Chief Conservator of Forests |
| | Use of DST | Tool used to view risk and report to higher officials and ministers, assist with planning and implementation of field activity, resource and media management, to facilitate inter / intra departmental coordination | | Risk maps to guide tick control, locations of IEC hoardings, monkey death surveillance, staff protection |
| | Data entry | Resource inventory at State level, checking data entered at lower levels | | |
| District Level | User | Deputy Director Animal Health | District Health Officer | Divisional Forest Officer |
| | Use of DST | Human case and monkey death locations and risk maps would facilitate supervision/planning of field implementation and tick vector control measures | Use tool to supervise field implementation, resource planning, manage media and mobilize local elected representatives | As above |
| | Data entry | District inventory of resources for PPE and tick control measures | District inventory of resources such as vaccines and repellents, check data from lower levels | Check data from lower levels |
| Taluk Level | User | Assistant Director Animal Health | Taluka Health Officer | Assistant Conservator of Forests |
| | Use of DST | Human case and monkey death locations and risk maps would facilitate supervision/planning of field implementation and tick vector control measures | Use tool to supervise field implementation, resource planning, manage media and mobilize local elected representatives | As above |
| | Data entry | Taluka inventory of resources for PPE and tick control measures | Taluka resource inventory and vaccine coverage, case locations, referrals, information on samples | Check data from lower levels |
| Ground level or Periphery | User | Veterinary Officers (Veterinary Dispensary) | Primary Health Centre Medical Officers (PHC MOs) | Range Forest Officer |
| | Use of DST | Human case and monkey death locations and post-mortem protocols in tool would facilitate role in tick vector control (on livestock), monkey post-mortems and sample collection | PHC MOs would use risk maps and IEC (e.g. on KFD symptoms, risks from ticks, tick prevention measures), stock inventory at Taluka level to plan and supervise field implementation | As above |
| | Data entry | post-mortem results and some monkey death locations (where not entered by the forest department) | PHC resource inventory, vaccine coverage of population, case locations and patient histories, information on suspected samples for the laboratory and referrals | Monkey death events, human case locations, vaccination status of staff, hoarding locations |
| | User | | Junior Health Workers | Forest Guard |
| | Use of DST | | Would use risk maps and IEC (e.g. on KFD symptoms, risks from ticks, tick prevention measures), stock levels of oil and vaccines, to target preventative measures in community | Raise awareness of risk areas and monkey deaths among public |
| | Data entry | | Fever surveillance, monkey deaths, human cases | Report monkey death events, human case locations |
| | User | | Accredited Social Health Activists (ASHAs) | |
| | Use of DST | | Would use risk maps and IEC (e.g. on KFD symptoms, risks from ticks, tick prevention measures), stock levels of oil and vaccines, to target preventative measures in community | |
| | Data entry | | Fever surveillance, monkey deaths, human cases | |

IEC = Information-Education-Communication.

presents the potential ways they would use the tool given their role in KFD management, and indicates whether and what type of data they might enter.

Key uses identified for mapped information in the DST at the ground-level were to enable targeting of field management activities such as community engagement on preventative measures, tick control and monkey death reporting. Awareness raising information on KFD symptoms, risks from ticks, tick prevention measures, monkey post-mortem protocols was also felt to be very beneficial at the ground-level. Ground-level users (i.e. field health workers linked to primary health centres) were likely to be able to enter important geographical and individual-level data (such as human case locations and case histories/outcomes, vaccine coverage, monkey death events and locations of tick and monkey samples testing positive for KFD virus (KFDV), forest guard and tourist routes) into the tool for use in management by the levels above. The risk maps that predict where human cases of KFD are likely to occur (based on whether locations have similar landscape conditions to locations that had KFD in the past) was one of the most popular functional aspect of the DST amongst stakeholders (Fig 3). It was suggested that preparedness for KFD could be enhanced if updated risk and outbreak maps could be provided each year, in time to guide pre-season tick surveillance, vaccination campaigns, as well as raising awareness amongst Primary Health Centres of KFD risk and diagnostic methodologies. Stakeholders suggested that inventories of public and animal health resources could be compiled within the DST to facilitate resource planning from Taluka level upwards, though

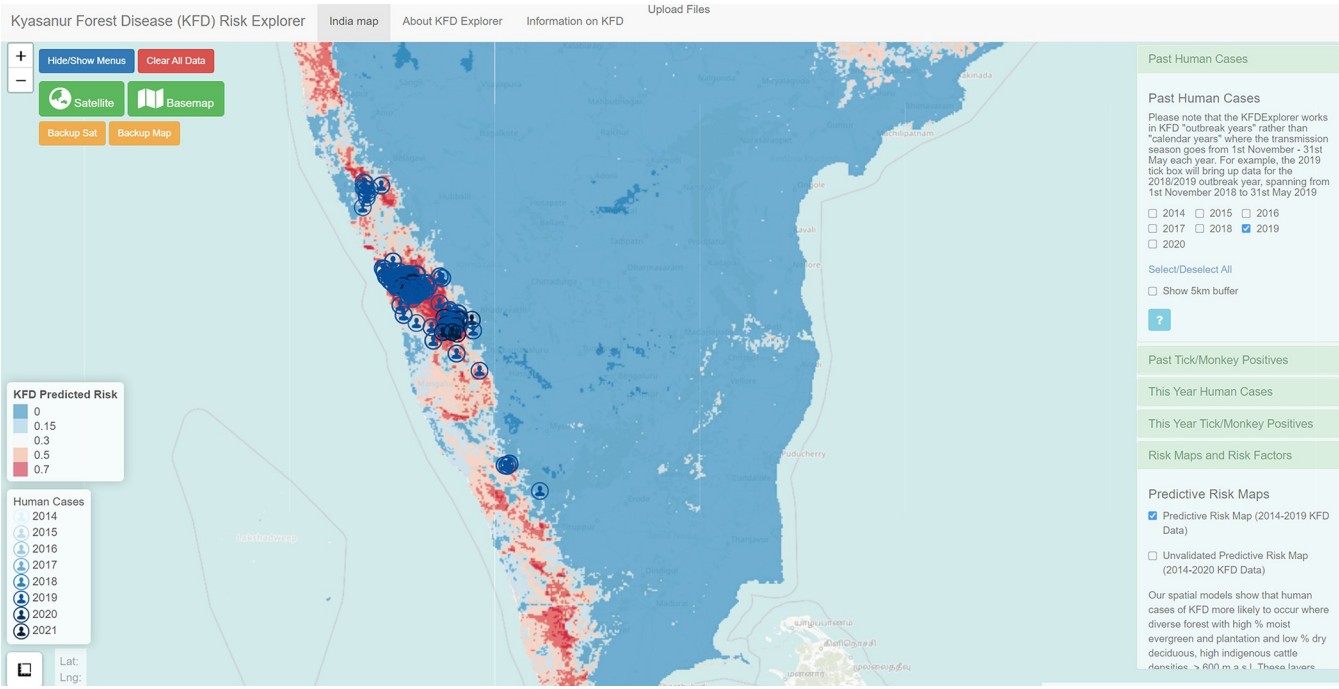

**Fig 3. Snapshot of the KFDExplorer Tool showing south India overlaid with the human cases reported in 2019 to the Department of Health and Family Welfare Services, Karnataka.** The areas predicted to be highly suitable for spill-over of KFD to humans are highlighted in red versus areas predicted to be unsuitable for spill-over in blue. The right hand menus in green indicate how data on environmental risk factors, on KFDV-positive dead monkeys and KFDV-positive ticks, can be visualised alongside human case locations, whilst the use of a detailed base map depicts landscape contextual features that guide management such as villages and roads. Source data: Map base layer is from the OpenStreetMap (https://wiki.openstreetmap.org/wiki/Standard_tile_layer). This dataset is available under a CC0 1.0 Universal (CC0 1.0) Public Domain Dedication license (https://creativecommons.org/publicdomain/zero/1.0/) and any copy of or work based on this dataset requires the following attribution: This dataset is based on the dataset produced by the OpenStreetMap Foundation (https://osmfoundation.org/). The administrative boundary dataset used in this figure is from HindustanTimesLabs (https://github.com/HindustanTimesLabs/shapefiles/), reproduced under the MIT License. Human case data are from the Department of Health and Family Welfare Services, Government of Karnataka.

this functionality is not yet incorporated and is planned to some degree under the Integrated Health Information Platform for India.

Other potential beneficiaries of the DST identified by stakeholders (and which the project team had not considered previously) included the Agriculture and Horticulture Departments, Revenue Department, the Education Department and the Tourism Department. This cohort of latent stakeholders were envisioned as having an important "stake" in contributing to holistic efforts to KFD prevention and control [63]. For example, it was suggested that the Tourism Department might use the risk maps to better prepare in the tourist areas at risk from KFD.

In terms of communication and inter-sectoral coordination, stakeholders felt that the DST may facilitate networking and linkage of data between departments and districts–a key aim of the OneHealth approach. Stakeholders further suggested that information from the DST could help with media engagement and mobilization of local elected representatives at Taluka and district levels and could feed into multi-level inter-sectoral meetings at village, Taluka, district, state and inter-state levels.

The workshop resulted in a project team decision to seek translation funding to build the Mobile Phone App and continue to scope out functionality and data flows with key beneficiaries in Karnataka, working through the Virus Diagnostic Laboratory, Shimoga and connecting with teams developing other information systems. The process of Mobile App development would be informed by piloting of the desk-based DST ("KFDExplorer tool") over the next 1–2 transmission seasons.

Another key outcome of the second workshop was the need to develop complementarity with existing tools. The World Health Organisation (WHO) Regional Office and the Indian Government have together developed and rolled out the Integrated Health Information Platform (IHIP) across India. Stakeholders felt that it was important that the DST be eventually integrated into IHIP to avoid peripheral workers from having to learn and enter data into multiple platforms. However, they also appreciated that the project DST was tailored for Kyasanur Forest Disease and, unlike IHIP at present, could be used to integrate and view information across sectors and included risk mapping and visualization of risk factors for KFD. Table 4 summarises some of the key distinctions between the MonkeyFeverRisk DST and the Integrated Health Information System that may affect and inform the integration process. Based on discussions with the regional office of the WHO after the workshop, a decision was made to continue developing the MonkeyFeverRisk DST (and viewed across sectors) as a stand-

**Table 4. Some key distinctions between the MonkeyFeverRisk DST and the Integrated Health Information System that may affect and inform the integration process.**

| | MonkeyFeverRisk DST | Integrated Health Information System |
|---|---|---|
| Spatial scale | Household level for case locations, which enables fine-scale linkages between the environment and human cases to be made | Health Centre and village level |
| Temporal scale | Long term risk area predictions for preparedness | Real-time event reporting and cluster identification |
| Data entry, access and visualization | Ideally, all can see some geographical data at all scales though requires state level agreements across sectors and careful protection of individual data | Set up so that users can only see the data for their own jurisdiction, not for neighboring areas or broader geographical areas (e.g. Primary Health Centre Medical Officers cannot see Taluk or block level data) Data sharing across states can be arranged by agreement of state level department heads |
| Sectoral involvement | Aiming for data entry and viewing across sectors though requires state level agreements across sectors and careful protection of individual data | Public Health Sector only can view at present Data sharing across sectors could be arranged by agreement of state level department heads |
| Risk factors | Overlay of risk factors e.g. forest types, tick and monkey positives, human populations | None |
| Risk maps | | Distance-based clustering in real-time during outbreak season |

alone tool, but that efforts would be initiated to integrate key data from the tool such as the risk maps, risk factors and data gathered from other sectors into IHIP. Similarly, Public Health data gathered by IHIP may feed into future versions of the MonkeyFeverRisk DST. The strategy thus was to roll out MonkeyFeverRisk DST first independently of IHIP on Indian Government servers and then make arrangements with the WHO IHIP team for further integration, taking account of feedback from a broader range of end users during the KFD season. As at the time of submission of this paper, the first step has been delayed by the COVID-19 pandemic and the need for the e-hospital to manage COVID-19 data and provide tools to inform the response.

The second workshop continued the co-production process started in the framing workshop. The process focused on knowledge integration, through the presentation of the work carried out in the project so far, and the development of the DST, based on the needs and constraints of the stakeholders using the DST. As a result of the workshop:

- The input from workshop participants on the decision-support tool fed into the further development of the tool.

- The input from workshop participants on the App was shared and discussed with an App developer. This is resulting in an App that could be shared with workshop participants.

- The project website was updated to provide more guidance documents such as maps, community guidance materials, resources for tick taxonomy and for members to share resources with each other.

- IHIP integration was explored and results shared with workshop participants.

## 4. Discussion

Despite the widespread and growing interest in the role of co-production of knowledge in bridging the gap between scientific evidence and policy implementation as a pathway to strengthen health systems [64,65], there remains uncertainty on what co-production of research entails, and how and when to implement it to maximise impact of research outputs [59,64,66]. This is especially true in the context of OneHealth operationalisation in LMICs, where there is relative dearth of empirical evidence on how to actively and meaningfully engage diverse range of stakeholders at the various stages of the research process to co-produce and mobilise contextually and policy-relevant knowledge [13,66,67]. In this study, we presented our reflections of various activities that comprised a co-production process to actively engage a range of stakeholders at different stages of our research to co-produce knowledge and develop a spatial decision support tool ("KFDExplorer tool") for improved KFD surveillance and control in India. As evidenced in our Theory of Change (see Fig 2), the workshops and engagement of key sectoral actors throughout the project duration framed the project's design, and resulted in improved identification of risk factors and knowledge of socio-ecological processes affecting management of KFD as well as indirect input into the KFDExplorer tool and risk model development.

Whilst, as highlighted in the wider literature and also experienced in our presented study, co-production in OneHealth is a difficult and resource-intensive process requiring regular iterative adjustments and flexibility, the beneficial outcomes justify its adoption [59,67,68]. Implementing meaningful co-production in the context of OneHealth with the aim of improving zoonotic disease management necessitates a nuanced understanding of the underlying contextual factors that shape outcomes of collaborative engagements between researchers and decision-makers [13,69,70]. Our findings contributes to the burgeoning OneHealth literature,

in which empirical studies are still sparse that describe successful and unsuccessful attempts to foster and sustain engagement with diverse stakeholders in co-producing OneHealth research to support zoonoses management, particularly in LMIC settings [56,64,67,71].

In our research project, the co-production process happened at the scale of the research team itself and led to impacts beyond our expectations. The development of the decision support tool benefitted from the pro-active steering by a member of the project team responsible for disease control in one state, and from having postdocs embedded in the Department of Health & Family Welfare Services (DHFWS), which led to improvements in the way surveillance data were collected and analysed. Good knowledge of the key stakeholder groups and embedding champions at the beginning part of the engagement process is pivotal for successfully creating co-produced research and translation into practice [59,72]. Moreover, trust building and continuous interactions between researchers and stakeholders, particularly in contexts where sectoral actors traditionally operate in "silos" remain fundamental [13]. Khayatzadeh-Mahani et al. [59] and Tembo et al. [72] argue that early engagement is a key principle for good co-production practice as it affords a window of opportunity to better understand stakeholders and 'gate-keepers' (values, interests, and context), develop trust and supports ethical and contextually-relevant research. Consistent with this argument, our flexible and early engagement with the DHFWS for instance, proved pivotal in establishing a good communication channel for receiving continuous semi-structured feedback (in addition to the more structured consensus-building NGT technique) and suggestions for tailoring project outputs for wider dissemination and uptake [59].

## 4.1 Creation of an interdisciplinary network

Consistent with the wider OneHealth literature [4,65,73], our findings suggest that fostering clear channels of communication among stakeholder groups at all stages of the co-production process is a critical first step towards improved collaborative arrangements. In this context, stakeholders themselves suggested the creation of an inter-disciplinary network (a dedicated and growing WhatsApp network sharing information relevant to KFD) which has facilitated information sharing and knowledge development in a quick and timely manner [55]. The workshops and collaborative working have also led to a broader culture of inter-sectorality, as demonstrated by the stakeholders in the second workshop highlighting wide-ranging sectors, scales and means required to leverage surveillance capacity across sectors (e.g. monkey deaths). For KFD, our workshops and co-production process, one of the first, enabled the formation of a large, interacting network of governmental and non-governmental stakeholders as well as representatives across sectors. In addition to a growing respect of each other's contributions (which were hitherto restricted to specific silos) [13,74], this informal network, with its positive interpersonal interactions, is likely to serve as a core group in future endeavours related to disease management and control.

## 4.2 Institution of culture of inter-sectorality

It is a common-placed view that better appreciation of the complexity and diversity of interests, values and context of stakeholders is necessary to overcome barriers and leverage opportunities for meaningful engagement of cross-sectoral stakeholders [64,70,75]. In this vein, the broader culture of inter-sectorality achieved over the course of our project facilitated a better understanding of different sectors, including their constraints. One example of this was the need to integrate public health constraints into DSTs. Stakeholders' discussion about resource inventories for KFD management revealed competing/conflicting needs of other diseases and/health conditions (e.g. key KFD personnel reassigned to support local COVID-19 response),

highlighting that health systems do not respond to individual zoonotic diseases in isolation [76]. Communities vulnerable to KFD are also likely to be affected by non-communicable diseases and other zoonotic infectious diseases such as scrub typhus and Leptospirosis [77,78]. Health system responses to co-occurring diseases always trade off against each other but this will be even more pronounced in forest-dependent communities (most affected by KFD) that are often remote from healthcare infrastructure [79]. Joint landscape-scale modelling and empirical studies of diseases offer the potential to understand how and where diseases are co-circulating and why they bundle together in the landscape [76].

Another benefit from inter-sectoral work was the improved understanding of the ethics of delivering risk models to policy makers (after Boden & McKendrick [80]. The co-production process allowed us to ascertain what information managers were already using to guide targeting of interventions and to reflect on whether the risk models provided would constitute an improvement against this baseline and how these models should be used operationally, given uncertainties, to avoid mis-direction of resources. In addition, the co-production process highlighted the need for data protection and data sharing agreements across sectors and levels, issues of equity of data sharing and entry, and clarity of roles of different beneficiaries in disease and data management.

## 4.3 Transparency between researchers and cross-sectoral stakeholders

A major challenge in generating policy-relevant evidence and context-specific solutions concerns incentivising and sustaining stakeholder interest throughout the research process [67]. Within this purview, the implementation of the co-production process facilitated full transparency with stakeholders about the modelling process, inputs, outputs and quality by establishing early and clear lines of communication and mutual understanding around available data and key risk factors. Validation and beneficence of our models was further facilitated by knowledge integration, in particular gathering and analysing outbreak data in partnership with health departments, during ongoing outbreaks [61]. Co-production was vital for gathering outbreak data that reflected locations of exposure in the landscape; better understanding contextual socio-ecological risk factors; and tailoring the spatial resolution and outputs to the scale of forest use, and public health interventions. It was an advantage to have researchers from our project embedded in the health department for coordination of multi-sectoral participation in process, for cross-sectoral linkage of data and priorities and to enable us to identify a mechanism by which ownership of the KFDExplorer tool could be successfully transferred to and sustained within the health system. Such a collaboration was made possible by the early co-production process used in this project. Our findings thus support the emerging evidence that suggest that, iterative and transparent engagement with stakeholders help circumvent contextual and institutional barriers and creates incentives for sustained stakeholder interest in the engagement process [72].

## 4.4 Improved preparedness for future outbreaks

As a result of the co-production process, the DST developed allows for a greater degree of preparedness for future outbreaks, including identifying knowledge gaps. The integration of knowledge from different sectors allowed for the identification of priority data gaps that impact management, for example on geographical variation in public health coverage and therefore vaccination coverage.

In terms of future co-production of zoonotic disease tools and management, a key aspect to consider is the influence of power dynamics between sectors on the co-production process [13,67,72]. As Agyepong et al. [67] argues, unequal power relations can weaken co-production

efforts and have negative ramifications, such as overlooking diverse forms of knowledge and perspectives. In addition, policy visibility of KFD changed over the process with events like large outbreaks in the Sagara district causing increasing interest and ownership from the public health sector at the expense of participation by the other sectors, whilst the outbreak of COVID-19 then redirected attention away from KFD. Whilst these externalities cannot necessarily be planned for, flexibility within the co-production process needs to be built in to allow for the 'normal' changes that are inherent to co-production processes (as was evident from the framing workshop), as well as the less common externalities. As part of the process, the project was able to adapt to input from stakeholders. A final consideration for future co-production processes is the need to manage expectations. As we noticed from the workshops, wider participation across sectors led to a larger "wish list" from a decision-support tool. This led to repeated attempts during and after workshops to manage expectations and time frames for the tool development. Such considerations need to be thought through and acted upon to avoid frustration from stakeholders and disengagement from the co-production process.

Outstanding questions, that are being addressed through a longer-term experimentation process, are whether the co-produced DSTs actually enhance disease preparedness in terms of human cases avoided due to improved targeting of interventions; whether the cross-sectoral engagement and ownership of the tool will be sustained; whether the tool can be fully embedded in health information systems and extended to other zoonotic diseases affecting forest communities. These notwithstanding, the foregoing has underscored the value of co-production approaches and how they could be highly applicable for operationalising OneHealth across different socio-spatial contexts.

## Supporting information

**S1 Fig. KFDExplorer Tool data flowchart showing the data flow processes to inform KFD surveillance and management.**
(TIF)

**S1 Table. The characteristics of the workshop participants.**
(DOCX)

**S2 Table. Prioritisation of KFD risk factors identified by stakeholders and what changed or was strengthened over the co-production workshops.**
(DOCX)

**S1 Text. Self-completion questionnaire for Workshop participants.**
(DOCX)

## Acknowledgments

We are extremely grateful to the stakeholders from across the Animal Health, Human Health, Forestry and Social Welfare sectors, community groups in India who gave up their valuable time to share their knowledge and experience with the project team. We appreciate the entire MonkeyFeverRisk project team. In particular, we acknowledge Drs F. Gerard, N. Balakrishnan and S. Bhat for their kind assistance with facilitation during the workshop and Dr G. Ainsworth for assistance with stakeholder mapping and workshop planning.

## Author Contributions

**Conceptualization:** Festus A. Asaaga, Prashanth N. Srinivas, Tanya Seshadri, Meera A. Oommen, Gudadappa S. Kasabi, Abi T. Vanak, Bethan V. Purse.

**Data curation:** Festus A. Asaaga, Juliette C. Young, Tanya Seshadri, Meera A. Oommen, Mujeeb Rahman, Shivani K. Kiran, Gudadappa S. Kasabi, Darshan Narayanaswamy, Abi T. Vanak, Bethan V. Purse.

**Formal analysis:** Festus A. Asaaga, Juliette C. Young, Prashanth N. Srinivas, Tanya Seshadri, Bethan V. Purse.

**Funding acquisition:** Juliette C. Young, Prashanth N. Srinivas, Meera A. Oommen, Gudadappa S. Kasabi, Sarah J. Burthe, Mudassar M. Chanda, Subhash L. Hoti, Abi T. Vanak, Bethan V. Purse.

**Investigation:** Festus A. Asaaga, Juliette C. Young, Prashanth N. Srinivas, Tanya Seshadri, Meera A. Oommen, Mujeeb Rahman, Darshan Narayanaswamy, Stefanie M. Schäfer, Sarah J. Burthe, Mudassar M. Chanda, Subhash L. Hoti, Abi T. Vanak, Bethan V. Purse.

**Methodology:** Festus A. Asaaga, Juliette C. Young, Prashanth N. Srinivas, Tanya Seshadri, Meera A. Oommen, Mujeeb Rahman, Shivani K. Kiran, Darshan Narayanaswamy, Stefanie M. Schäfer, Sarah J. Burthe, Mudassar M. Chanda, Subhash L. Hoti, Abi T. Vanak, Bethan V. Purse.

**Project administration:** Bethan V. Purse.

**Supervision:** Juliette C. Young.

**Validation:** Juliette C. Young, Shivani K. Kiran, Darshan Narayanaswamy, Tom August, Mark Logie, Abi T. Vanak, Bethan V. Purse.

**Visualization:** Festus A. Asaaga, Juliette C. Young, Darshan Narayanaswamy, Tom August, Mark Logie, Bethan V. Purse.

**Writing – original draft:** Festus A. Asaaga, Juliette C. Young, Bethan V. Purse.

**Writing – review & editing:** Festus A. Asaaga, Juliette C. Young, Prashanth N. Srinivas, Tanya Seshadri, Meera A. Oommen, Mujeeb Rahman, Shivani K. Kiran, Gudadappa S. Kasabi, Darshan Narayanaswamy, Stefanie M. Schäfer, Sarah J. Burthe, Tom August, Mark Logie, Mudassar M. Chanda, Subhash L. Hoti, Abi T. Vanak, Bethan V. Purse.

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
