## [Decision Letter · Decision Letter 0]

4 Sep 2021

PGPH-D-21-00480

Co-production of knowledge as part of a OneHealth approach to better control zoonotic diseases.

Dear Dr. Asaaga,

Thank you for submitting your manuscript to PLOS Global Public Health. After careful consideration, we feel that it has merit but does not fully meet PLOS Global Public Health’s publication criteria as it currently stands. Therefore, we invite you to submit a revised version of the manuscript that addresses the points raised during the review process.

We look forward to receiving your revised manuscript.

Kind regards,

Giridhara Rathnaiah Babu, MBBS, MBA, MPH (Epidemiology), PhD (Epidemiology)

Academic Editor

Journal Requirements:

Additional Editor Comments (if provided):

At the outset, I take this opportunity to thank you personally, and all the reviewers recognize your hard work and motivation to charter in publishing the interdisciplinary work. Two of the reviewers have expressed that this manuscript merits an opportunity to publish provided the report of the results is modified and the manuscript is revised. One of the reviewers has pointed to self-citations and adapting another figure from your published manuscript. Please make the modifications as suggested by the reviewer.

At the conceptual level, we recognize that your team has profound knowledge of One Health. The rationale for the co-production process conducted against the established systems approach in One Health needs to be explained. I will be happy to receive a revised manuscript addressing the specific recommendations by the reviewers.

Reviewers' comments:

Reviewer's Responses to Questions

**Comments to the Author**

1. Does this manuscript meet PLOS Global Public Health’s publication criteria? Is the manuscript technically sound, and do the data support the conclusions? The manuscript must describe methodologically and ethically rigorous research with conclusions that are appropriately drawn based on the data presented.

Reviewer #1: Yes

Reviewer #2: Yes

Reviewer #3: Yes

Reviewer #4: Yes

Reviewer #5: Partly

2. Has the statistical analysis been performed appropriately and rigorously?

Reviewer #1: No

Reviewer #2: N/A

Reviewer #3: No

Reviewer #4: N/A

Reviewer #5: N/A

3. Have the authors made all data underlying the findings in their manuscript fully available (please refer to the Data Availability Statement at the start of the manuscript PDF file)?

Reviewer #1: No

Reviewer #2: Yes

Reviewer #3: No

Reviewer #4: Yes

Reviewer #5: Yes

4. Is the manuscript presented in an intelligible fashion and written in standard English?

Reviewer #1: Yes

Reviewer #2: Yes

Reviewer #3: No

Reviewer #4: Yes

Reviewer #5: Yes

5. Review Comments to the Author

Reviewer #1: The co-production initiative that includes several stakeholders from public health, animal health and environment to create a DST would be very beneficial for the control of zoonotic diseases such as KFD. The information generated from the manuscript is highly relevant for future decision makers in One Heath. The authors did a great job in explaining the co-production process and how they could contribute to the future disease outbreak management.

However, the manuscript lack specific details on how the data were analyzed that resulted in the development of DST. Here are few suggestions for the authors

1. How were the stakeholders identified? Where they involved in policy/decision making process. It would be better to list out the title of the stake holders who attended both the workshops.

2. The authors should describe the approach used to develop desk-based spatial DST mentioned in line 219

3. The authors also need to give a detailed explanation on how the data generated from the two workshops were analyzed. For example how was the risk factors identified and how did they account of the unequal participation from different sectors (6 participants from public health vs 10 participants from animal health in Workshop 1)

4. What was the criteria used by the authors to either include or exclude the risk factors (line 278) and how was the users and uses identified by the stakeholders screened to select for the key potential users and uses (Table 3).

5. Is there any link for accessing the DST tool so that the reader could have a better understanding about the usefulness of the approach

Reviewer #2: It is a well written manuscript. There are a few punctuation errors that require attention. The introduction could be made more crisp and to the point.

In the manuscript as a whole, there are a few reiterations of things that have already been touched upon. The manuscript will benefit from an editing to make the entire thing seem more put-together and without any repetitions.

Kindly do the needful.

Reviewer #3: Title: If this co-production of knowledge is from a case study of southern India, then this should reflect on the title.

Abstract

L29: I could rather say, necessarily not working together. Although there might be some knowledge, and they may work under certain circumstances, this is not necessarily happening.

L32: reduction of disease impact or risk mitigation at the interface of human-animal-environment. The author may like to justify why they are saying the reduction of disease impact rather than risks.

L37: If the case study is about KFD, that must align with the L32 statement and reflect in the title.

Introduction

L55-56: It seems the sentence is not well placed.

L65-68: Difficult to generalize, as it strongly depends on the governance structure and the functionalities. Yes, it's true if it has been written from the Indian perspective, but authors need to specify here.

L74: Why ‘global’. One Health is local too. So why global One Health?

L77: The One Health sope is not just to reduce the disease impact. It must be valued as an integrated risk management perspective too.

L80-83: Which models authors are talking about? Need clarification

Overall well-written background; however, I have a gross critique. Why are authors introducing the concept of co-production here? In the rationale of the One Health approach, cross-sectoral collaboration is the base, and the theme lies with the systems approach. There must be a strong argument, how this co-production process is different from the systems approach of solving multi-dimensional issues, which is needed in the background.

Methods

L156-187: I could see multiple self-citations (please refer to other literature). Rather than writing about the whole disease management system here in the method section. It will be great to replace the KFD disease management system with a three-tier approach of central-state-local governance and all the stakeholders involved in this process.

Fig.1: It’s the same figure from author’s recently published article. Can I suggest developing a new framework specific to KFD, although the aim is to provide a generic system to readers?

L210-248: It is important how researchers invited these participants. In the participatory approach, it is an integral component as the outcome depends on who participated and the rank of the stakeholders, including experiences. Please explain and justify why there is a discrepancy in the number of stakeholders from three domains of One Health. Also, add a para on the pre-workshop-related development works.

Results

L271-284: Is there any decision-making tree analysis or hierarchical analysis conducted as part of the problem prioritization? If so, please specify. If not, then justify the process. And how the weightage calculation (if any) is conducted needs to be expanded. If not, then why there was no such process done, justify.

L303-341: Readers may like to have a brief idea of the KFD project outline prior to the workshop. And how that priority has changed over the co-production workshops.

L388-424: Some of the recommendations are aligned with the findings of this workshop. The authors may like to shift those recommendations (perhaps overall and specific) towards the end of the draft with a separate section.

Conclusion

Authors may like to provide a guidance framework in the form of process flow, including the impact matrix for better understanding. Also, link factors that could potentially affect each step of the process flow and minimize those.

Reviewer #4: This is an excellent article that makes a very timely contribution to our understanding of how to move the One Health approach from theory to practice. The structure of the paper is set out effectively and the overall argument supported well.

Suggestions for revisions:

In the methods section, there should be more information on how the data generated as part of the workshops was analyzed. Was it imported into Nvivo, or a similar qualitative research software? What type of analysis (thematic, constant comparative, discourse) was conducted? How were coding categories developed (inductively, deductively), etc?

It is interesting that stakeholders wanted to have the DST be integrated with WHO's IHIP but that this was not done. Could you provide more details on why.

Additional comments:

P.9 Fig.1 Capture should be shortened and acronyms placed at end of article.

P.22 line 329 Punctuation missing after the word direction

P.27 Fig.3 Capture should be shortened and figure explained in text/footnote not heading

Reviewer #5: General Comments:

The manuscript presents an interesting case study of a scientific project attempting to engage in a more inclusive process with stakeholders of their work. The reflective work done to prepare this manuscript is highly valuable and should be acknowledged. It is a known struggle for scientists accustomed to quantitative research to present qualitative work with the same rigor. The progress in inter- and transdisciplinary health sciences will require much more of these pioneering efforts. I therefore commend this work for publication after some thorough revision.

The article would benefit from some reorganisation according to the “standards for reporting qualitative research”. In particular, the result section should provide the data that the authors employ to draw their conclusions in the discussion section. As it stands, the article appears rather as an opinion piece than a scientific report from qualitative research. Also, the discussion should make substantially more reference to other work conducted in the field of one health and transdisciplinarity. Maybe, the authors could formulate a working hypothesis for the introduction of this work, which then would facilitate structuring the discussion.

Specific Comments:

L59: The link between ecosystem degradation and disease burden needs some broader references looking into the problem, e.g. highly relevant references would be

https://www.frontiersin.org/articles/10.3389/fvets.2021.661063/full ; https://doi.org/10.1016/j.biocon.2020.108707

L74-7: consider inverting the statement “...to endorse and facilitate health” instead of “…reduce the impact of diseases”. There are implications when focusing on diseases rather than health, particularly in regards to the stakeholders that are considered. When intending a one health approach, the objective should be positive rather than the absence of negative. For detailed rationale behind this, please see: https://link.springer.com/article/10.1007%2Fs11625-019-00674-z

L86-9: In the context of One Health, the authors could consider also referring to transdisciplinarity as a term used to describe knowledge co-production, e.g. http://www.ncbi.nlm.nih.gov/pubmed/20391395 ; http://link.springer.com/10.1007/s13280-012-0372-4 ; https://www.cabi.org/bookshop/book/9781789242577/

L102-4: Besides these pertinent aspects, the authors could consider mentioning that further aspects that require an integrated approach have been identified in the specific context of one health initiatives, namely improved equity, stewardship, and efficiency (https://www.frontiersin.org/articles/10.3389/fpubh.2017.00020/full)

L104-8: An important aspect that is missing here, is that citizens have tacit knowledge that must be mobilized for effective solutions. I.e. they have locally valid knowledge that is not scientifically recorded or available. In conventional approaches, such knowledge is often discovered throughout the project, but not included in the conception phase.

L 183: replace “actor” by “sector”.

L206: The inclusion/exclusion of stakeholders in a co-production process has fundamental impact on its outcomes. Hence it would be important here to outline the stakeholder management plan, including the identification, priorisation and engagement process.

L214: From the affiliation mentioned above, it seems that no citizen representatives were engaged. This may seriously hamper the resulting conclusions in regard to their applicability. This should be discussed in the limitations of this study.

L220: With “spillover”, usually events are described that are singular changes of host species, i.e. when SARS-CoV2 adapted from the intermediate host to humans, or NIPAH from pigs to humans. In the present case, the authors should refer to probability maps for human infections. A more detailed explanation of the differences can be found here: https://www.frontiersin.org/articles/10.3389/fpubh.2020.596944/full

L223: It might be of interest to the readers, how the authors handled intellectual property rights at this point. "Allowing" implies that this would have been prohibited otherwise.

L234: maybe state the objectives or provide a chart of the various steps in the co-production process with the corresponding stakeholders. E.g. it seems that the DST was developed in the lab by the authors, and then exposed to the assumed end-users for validation. It would be nice to see how the workshop1 information was integrated in the DST models.

L240: consider revising the verb “inputted”

L241: I suggest revising this sentence: "The workshop relied entirely on participation using a range of different methods including group discussions, brainstorming and priorization exercises."

The "approach" refers to the "attitude of the researchers" or "the more fundamental paradigm", while what they are describing are participatory methods. Whether these are engaging or not (qualification of the participation) essentially depends on the facilitators, thus the authors would have to declare the criteria for their self-assessment to declare if participation was high.

L244-6: What qualifies Fig 2 as a snapshot? As I understand, it is a graphical representation of the retrospective theory of change. Please omit “framework”. Please refine the process according to the comment above - or use a different figure.

L269: Consider using the heading "results". In my view, insights should be mentioned in the discussion section. In general please consider following the relevant guidelines from the equator network, e.g. the standards for reporting qualitative research:

https://www.equator-network.org/reporting-guidelines/srqr/

L280: It is interesting to read that the modelling approach (spatial risk models) seems to have been pre-defined by the authors. Implicitly this excludes much information that is eminently important for early warning, such as the monkey deaths. I would like to read some reflection on the choice of model and its impact on the ability to respond to the stakeholder expectations. Alternatives would have been e.g. Fuzzy cognitive maps or agent based modelling. For a more comprehensive list of possible participatory modelling approaches please refer to Voinov, 2018: https://doi.org/10.1016/j.envsoft.2018.08.028

L337: I assume, the authors mean "resolution" with “grain”.

L362: It is not quite clear to me, what ground-level means in this context and whether it is contrasted or synonym to the peripheral user, i.e. a field collaborator. Please specify the distinctions. It feels as if the authors were using a vertical and a concentric hierarchical image to express the same.

L370: This rationale ignores the real time information about monkey deaths, which is highly relevant. The implemented algorithm endorses a heuristic approach, which doesn't necessarily reflect the true probability of infection, i.e. it is excessively conservative and identifies high probabilities, where they may not occur anymore, while they'd ignore evidence for high probabilities, where no human cases have occurred.

Secondly the authors imply that the probability of infection equals the risk. However, in health sciences risk is usually the product of likelihood and consequences. In this case, the consequences would strongly depend on the subject being exposed to a certain likelihood. I suggest, the authors use “probability maps” or landscapes rather than "risk maps" as a term.

L388: The authors could reflect whether the users they identified are actually stakeholders or not potentially shareholders in the problem. The distinction has nicely been elaborated in "Systemic Decision Making" by Hester and Adams (2017, Springer). Usually, the agents responsible for the management are not suffering from the consequences of poor management and consequently are share- rather than stakeholders in regard to their motivation(s) to act. This is quite relevant when it comes to the stewardship generated by integrative projects. The issue was illustrated by an evaluation of previous one health projects including one from Kerala on KFD: https://www.frontiersin.org/articles/10.3389/fpubh.2021.653398/full

L401-4: This belongs in the methods

L442 ff: The authors don't seem to be aware of the various efforts that have previously been made to learn from one health initiatives and other transdisciplinary work. A starting point would be the Hitziger 2021 paper, but multiple others have been published and should be used to compare the conclusions drawn by the authors here.

In my view this discussion lacks references and reflection. While I acknowledge the self-comparison, I would highly recommend some external comparisons which would also allow to elaborate what was particularly exemplary in this project over other similar endeavors. e.g. http://journals.sagepub.com/doi/10.1177/160940690700600401;
http://link.springer.com/10.1007/s11625-015-0334-4;
http://bmcpublichealth.biomedcentral.com/articles/10.1186/1471-2458-13-897;
https://doi.org/10.1016/j.evalprogplan.2021.101991;

https://www.frontiersin.org/research-topics/5479/;

L466: Given the precedent project evaluated in Hitziger 2021, I don't think this claim “the first of its kind” is appropriate.

Figures:

Fig2:

Please also indicate how "culture of inter-sectorality, knowledge integration and preparedness were assessed (all other outcomes/ impacts have a criterion).

Fig3: please replace “spillover” by “human infections”

Tables:

Table 1 and 2: please replace “spillover” by “human infections”

6. PLOS authors have the option to publish the peer review history of their article (what does this mean?). If published, this will include your full peer review and any attached files.

**Do you want your identity to be public for this peer review?** For information about this choice, including consent withdrawal, please see our Privacy Policy.

Reviewer #1: No

Reviewer #2: **Yes: **Fayiqa Ahamed Bahkir

Reviewer #3: **Yes: **Sandul Yasobant

Reviewer #4: No

Reviewer #5: **Yes: **Simon R. Rüegg

---

## [Decision Letter · Decision Letter 1]

9 Jan 2022

Co-production of knowledge as part of a OneHealth approach to better control zoonotic diseases.

PGPH-D-21-00480R1

Dear Dr. Asaaga,

We're pleased to inform you that your manuscript has been judged scientifically suitable for publication and will be formally accepted for publication once it meets all outstanding technical requirements.

Within one week, you'll receive an e-mail detailing the required amendments. When these have been addressed, you'll receive a formal acceptance letter and your manuscript will be scheduled for publication.

An invoice for payment will follow shortly after the formal acceptance. To ensure an efficient process, please log into Editorial Manager at https://www.editorialmanager.com/pgph/ click the 'Update My Information' link at the top of the page, and double check that your user information is up-to-date. If you have any billing related questions, please contact our Author Billing department directly at authorbilling@plos.org.

Kind regards,

Giridhara Rathnaiah Babu, MBBS, MPH, PhD

Academic Editor

Additional Editor Comments (optional):

Reviewers' comments:

Reviewer's Responses to Questions

**Comments to the Author**

1. If the authors have adequately addressed your comments raised in a previous round of review and you feel that this manuscript is now acceptable for publication, you may indicate that here to bypass the “Comments to the Author” section, enter your conflict of interest statement in the “Confidential to Editor” section, and submit your "Accept" recommendation.

Reviewer #1: All comments have been addressed

Reviewer #2: All comments have been addressed

Reviewer #5: All comments have been addressed

2. Does this manuscript meet PLOS Global Public Health’s publication criteria? Is the manuscript technically sound, and do the data support the conclusions? The manuscript must describe methodologically and ethically rigorous research with conclusions that are appropriately drawn based on the data presented.

Reviewer #1: Yes

Reviewer #2: Yes

Reviewer #5: Yes

3. Has the statistical analysis been performed appropriately and rigorously?

Reviewer #1: Yes

Reviewer #2: Yes

Reviewer #5: N/A

4. Have the authors made all data underlying the findings in their manuscript fully available (please refer to the Data Availability Statement at the start of the manuscript PDF file)?

Reviewer #1: Yes

Reviewer #2: Yes

Reviewer #5: Yes

5. Is the manuscript presented in an intelligible fashion and written in standard English?

Reviewer #1: Yes

Reviewer #2: Yes

Reviewer #5: Yes

6. Review Comments to the Author

Reviewer #1: The authors did an excellent job in including relevant reviewer comments. Detailed description of methods utilized and data analysis and the limitations was very useful. Overall, the materials and methods reads well, however, introduction and discussion sections were pretty long and could be made more simple by avoiding redundancies.

Reviewer #2: The article is well-written, but there a few corrections to be made.

Introduction too long, please make it crisp and to the point.

Few spelling mistakes in the manuscript, kindly go through manually, as running an automated spell check might not pick up irrelevant words, examples as taken from the script below:

2.1.1 Actors involved in disease management in India

(line 214)exercise, which culminated in a list of current and potential cross-sectoral actors involved in

Reviewer #5: Congratulations to the authors. The discussion has greatly improved and exploited the potential of this work.

7. PLOS authors have the option to publish the peer review history of their article (what does this mean?). If published, this will include your full peer review and any attached files.

**Do you want your identity to be public for this peer review?** For information about this choice, including consent withdrawal, please see our Privacy Policy.

Reviewer #1: No

Reviewer #2: **Yes: **Fayiqa Ahamed Bahkir

Reviewer #5: **Yes: **Simon R. Rüegg, Section of Epidemiology, Vetsuisse Faculty, University of Zürich, Switzerland
